# TALEN outperforms Cas9 in editing heterochromatin target sites

Surbhi Jain[1,9], Saurabh Shukla [2,3,9], Che Yang[1], Meng Zhang [2], Zia Fatma[2,4], Manasi Lingamaneni[1], Shireen Abesteh[1], Stephan Thomas Lane[4], Xiong Xiong[2], Yuchuan Wang[5], Charles M. Schroeder[2,6,7], Paul R. Selvin[3,7,8] & Huimin Zhao [1,2,4,7✉]

Genome editing critically relies on selective recognition of target sites. However, despite recent progress, the underlying search mechanism of genome-editing proteins is not fully understood in the context of cellular chromatin environments. Here, we use single-molecule imaging in live cells to directly study the behavior of CRISPR/Cas9 and TALEN. Our single-molecule imaging of genome-editing proteins reveals that Cas9 is less efficient in heterochromatin than TALEN because Cas9 becomes encumbered by local searches on non-specific sites in these regions. We find up to a fivefold increase in editing efficiency for TALEN compared to Cas9 in heterochromatin regions. Overall, our results show that Cas9 and TALEN use a combination of 3-D and local searches to identify target sites, and the nanoscopic granularity of local search determines the editing outcomes of the genome-editing proteins. Taken together, our results suggest that TALEN is a more efficient gene-editing tool than Cas9 for applications in heterochromatin.

[1] Department of Biochemistry, School of Molecular and Cellular Biology, University of Illinois at Urbana—Champaign, Urbana, IL 61801, USA. [2] Department of Chemical and Biomolecular Engineering, University of Illinois at Urbana—Champaign, Urbana, IL 61801, USA. [3] Center for the Physics of Living Cells, University of Illinois at Urbana—Champaign, Urbana, IL 61801, USA. [4] Carl R. Woese Institute for Genomic Biology, University of Illinois at Urbana—Champaign, Urbana, IL 61801, USA. [5] Computational Biology Department, School of Computer Science, Carnegie Mellon University, Pittsburgh, PA 15213, USA. [6] Beckman Institute for Advanced Science and Technology, University of Illinois at Urbana-Champaign, Urbana, IL 61801, USA. [7] Center for Biophysics and Quantitative Biology, University of Illinois at Urbana—Champaign, Urbana, IL 61801, USA. [8] Department of Physics, University of Illinois at Urbana—Champaign, Urbana, IL 61801, USA. [9] These authors contributed equally: Surbhi Jain, Saurabh Shukla. ✉email: zhao5@illinois.edu

Clustered regularly interspaced short palindromic repeats (CRISPR)/CRISPR-associated protein 9 (Cas9) and transcription activator-like effector nuclease (TALEN) are programmable DNA search engines that query genomic sequences for target-specific editing[1]. Both Cas9 and TALEN can recognize a custom genetic sequence but have strikingly different mechanisms of target-site binding[2]. Cas9 can be programmed to find a specific DNA sequence upstream of an indispensable 3-nucleotide motif (protospacer adjacent motif or PAM) by designing a single guide RNA (sgRNA) that mediates target-site binding through DNA-RNA pairing[3]. On the other hand, the DNA-binding domain of a TALEN is comprised of a tandem array of 33–34 amino acid (aa)-long customizable monomers that theoretically can be assembled to recognize any genetic sequence following a one-repeat-binds-one-base-pair recognition code[4,5]. In vitro single-molecule studies have shown that TALEs (nuclease-free analogs of TALENs) utilize a unique rotationally decoupled, "molecular zip-line" mechanism for target-site search along DNA; it does this by translating along the DNA backbone without rotating or tracking the major groove[6,7]. However, it is not known how TALEs maneuver the complex nuclear architecture and search for the target-site in vivo. Previous studies have presented a conflicting view of the CRISPR/Cas9 search mechanism, often using dCas9 (a nuclease-deficient Cas9), either described as pure 3-D diffusion[8–10] or capable of 1-D diffusion along DNA[11].

In this work, we directly observe the search behavior of dCas9 and TALE proteins in different chromatin environments in vivo. By analyzing the trajectories of single protein molecules in live cells, we characterize the local search mechanisms of TALE and dCas9 in euchromatin and heterochromatin regions. Our results show that Cas9 is less efficient than TALEN in heterochromatin regions because Cas9 tends to become encumbered by local searches on non-specific sites. To further assess the functional implications of the differences in search behaviors, we conducted a TIDE (Tracking of Indels by Decomposition)[12] analysis of TALEN and Cas9, revealing that TALEN was up to fivefold more efficient than Cas9 in the constrained heterochromatin regions of the genome. Overall, this combined strategy allows us to independently investigate the search mechanism as well as the editing efficiency of both genome-editing proteins.

## Results

**Live-cell imaging of TALE and dCas9 proteins.** Live-cell single-molecule fluorescence microscopy[13,14] was used to directly observe the search dynamics of TALE and dCas9 proteins in mammalian cells. We designed a TALE protein that is primarily in the search mode because it has few binding sites—specifically the cystic fibrosis transmembrane conductance regulator (CFTR) genomic loci in euchromatin with less than 4 binding sites in the genome. We also synthesized a TALE protein with multiple target sites—in particular, a TALE targeting the Alu retrotransposon elements[15], with an estimated 1 million interspersed target sites (Fig. 1a). In both cases, the proteins were fused with a Halotag domain[16] and were constructed using an in-house liquid handling robotic system[17], enabling 1:1 stoichiometric labeling with JF 549 dye[18] (Supplementary Fig. 1). Similarly, we also designed guide-RNAs targeting CFTR and Alu sites to be used with dCas9 proteins. We performed control experiments with the core histone protein H2B (Histone 2B), a widely studied DNA-binding protein[19,20] (Supplementary Table 1).

**TALE and dCas9 exhibit two major search behaviors.** Protein search dynamics were analyzed using two different imaging conditions: short-exposure times (10–20 ms) to study fast

diffusion kinetics (Fig. 1b, Top) and long-exposure times (500 ms) to characterize residence times of the bound molecules (Fig. 1b, Bottom). First, we analyzed the data obtained for the short-exposure time imaging condition. The seminal theoretical framework described by Berg, Winter, and von Hippel identified that specific DNA-binding proteins undergo four major processes of translocation, namely, (i) "long-range" or "macroscopic" disassociation-reassociation events (3-D diffusion), (ii) "short-range" or "microscopic" disassociation-reassociation events (hopping), (iii) ring-closure or "intersegmental transfer" in case the protein has 2 DNA-binding sites and lastly (iv) "sliding" (1-D diffusion) along the DNA molecule[21]. More recently, 3-D diffusion and hopping have been referred to as global search, and 1-D sliding and jumping (<5 bp) have been characterized as local search[19,22]. Our experimental setup is unable to differentiate between a jump and a pure 1-D sliding translocation due to the lower bound of the short-exposure time and the inability to visualize DNA. Fast-diffusing molecules carry out global search, whereas slow-diffusing molecules are carrying out local search[19].

We performed multi-state Gaussian fitting on normalized diffusion coefficient histograms of TALE and dCas9 proteins (Fig. 1c). Diffusion histograms of CFTR-TALE exhibited two types of search behaviors, a "fast" diffusion (red curve) and a "slow" diffusion (green curve) with significant overlap (Fig. 1c, top image). These results show that TALE proteins are capable of switching (on a timescale of 20 ms) from fast to slow diffusion and vice versa. We posit that the fast-diffusing populations (red curve) result primarily from global search events such as hopping and 3-D diffusion, whereas the slow-diffusing populations (green curve) include locally searching molecules. In conclusion. the double peak behavior of the normalized diffusion coefficient histograms suggests that TALE proteins engage in global search as well as local search behavior. A similar analysis of diffusion coefficient histograms for H2B controls also showed evidence of two molecular populations captured by fast and slow-moving H2B molecules (Supplementary Fig. 2).

We further studied the search behavior of nuclease-deficient Cas9 (dCas9) in living cells. Our results show that dCas9 also exhibits two major search behaviors, similar to TALE. The characteristic "fast" and "slow" diffusion populations make up the CFTR-dCas9 diffusion processes (Fig. 1c, bottom). The kinetic parameters of the search process are comparable between CFTR-TALE and dCas9. We have found the target-search process of TALE molecules similar to that of dCas9 in a live-mammalian nucleus. However, it has been reported that Cas9 tends to outperform TALEN when editing sequences in the open chromatin[2]. In the next sections, we aimed to investigate the fundamental molecular differences of TALE and dCas9 target-search processes and how they may affect the editing outcomes.

**Long exposure time condition allows visualization of DNA-bound proteins.** We further imaged proteins using long-exposure times (500 ms), allowing for the visualization of DNA-bound molecules (bottom image, Fig. 1b). In this imaging condition, fast-moving proteins diffuse in the background, and only DNA-bound proteins are visible. Using long-exposure times, we determined the residence time histograms of bound TALE and Cas9 proteins that targeted CFTR and Alu genomic loci. Residence time histograms could not be fit with a single-exponential decay function (Supplementary Fig. 3). Histograms of Alu- and CFTR-TALE residence times were well described by a two-component exponential decay model, which suggests the presence of "non-specifically" and "specifically" bound molecules[23] (Fig. 2a). Binding times were determined for both populations

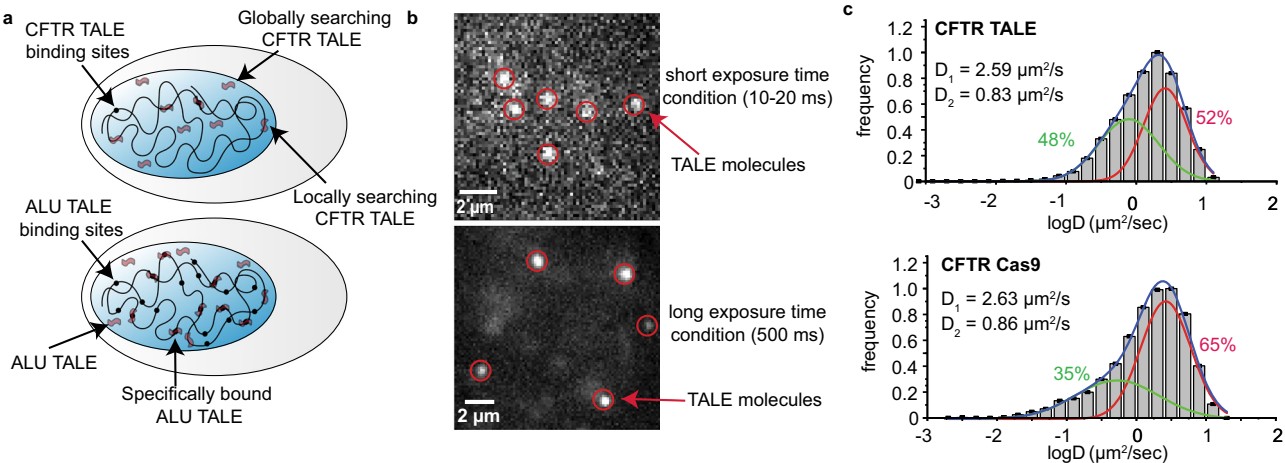

**Fig. 1 Single-molecule imaging of TALE target-search dynamics. a** Schematic of TALE-Halotag imaging. The difference in density of Alu- and CFTR-TALE binding sites is illustrated. Different modes of target search, i.e., global search and local search, are highlighted with arrows. **b** Microscope images of short-exposure time (top image) and long-exposure time condition (bottom image) are shown. Red circles denote protein molecules. **c** Diffusion coefficient histograms of TALE and dCas9 proteins targeting the CFTR elements in the human genome. There are two characteristic populations, fast and slow-moving, which are present in the case of both TALE and dCas9. $N_{CFTR\ TALE}$ = 16,348 and $N_{CFTR\ Cas9}$ = 38,395, where $N$ denotes the number of data points. Source data are provided as a Source Data file.

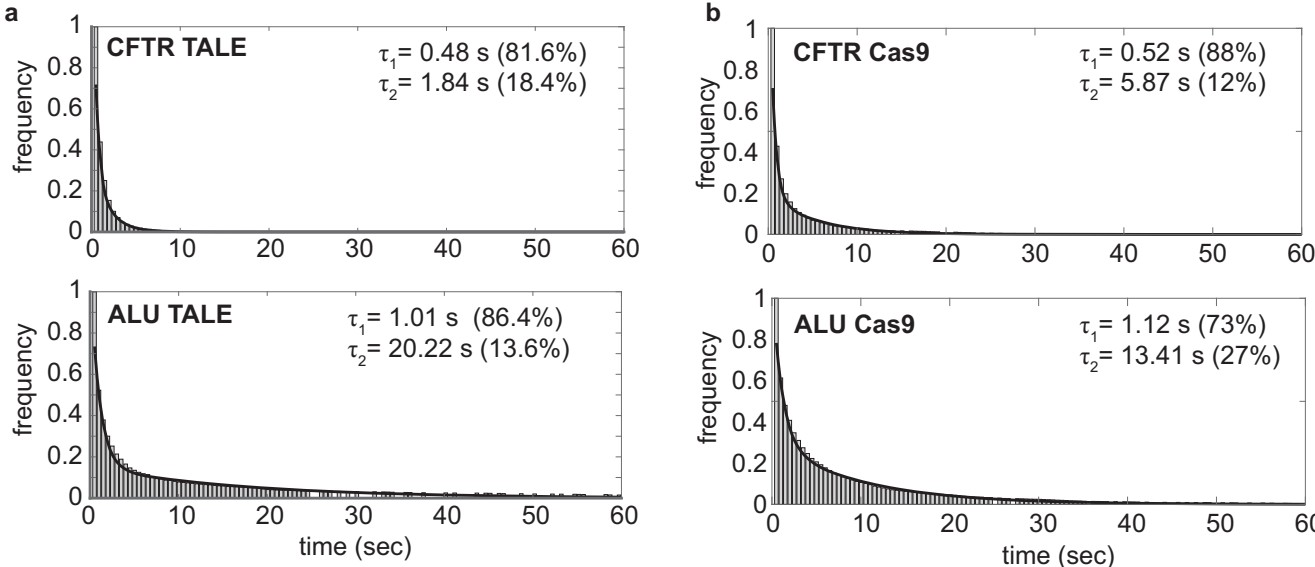

**Fig. 2 Residence times of TALE and dCas9 on chromatin. a** Residence times of TALE proteins in the large exposure time (500 ms) movies are plotted and fitted to the double-exponential decay function. Large exposure time movies only capture the molecules that are bound to the genome. The top and bottom figures show the residence times of CFTR TALE and Alu TALE, respectively. Plots also show the characteristic residence times of each of the two populations ($\tau_1$ and $\tau_2$) and the corresponding population percentages (in brackets). $N_{CFTR\ TALE}$=1373 and $N_{Alu\ TALE}$=1918, where $N$ represents the number of data points. **b** Residence time histograms of CFTR and Alu dCas9 are shown. Similar to TALE histograms, characteristic timescales of and corresponding populations of dCas9 proteins are also shown. $N_{CFTR\ Cas9}$= 2159 and $N_{Alu\ Cas9}$= 3638. Source data are provided as a Source Data file.

based on the double-exponential decay model, after correcting for photobleaching (see Methods). Our results show that short-lived populations of CFTR-TALE ($\tau_{1,\ CFTR}$) and Alu-TALE ($\tau_{1,\ Alu}$) have lifetimes of 0.48 s and 1.01 s, respectively (Fig. 2a). However, residence times of the long-lived population ($\tau_2$) differed significantly between CFTR-TALE and Alu-TALE, such that CFTR-TALE and Alu-TALE showed residence times of 1.8 s and 20.2 s, respectively. Because Alu TALE has more than a million target sites, the longer residence time $\tau_2$ for Alu-TALE reflects the behavior of proteins that are bound to specific target sites. Moreover, CFTR has only a few target sites (<4), such that the vast majority of CFTR TALE–DNA interactions are likely to be

non-specific. Therefore, we deduced that TALE spends an average of 1.8 s ($\tau_2$ of CFTR TALE) at non-specific sites and 20.2 s ($\tau_2$ of Alu-TALE) bound to target sites. The short-lived populations described by $\tau_1$ reflect the dynamics of the transitionary molecules, representing the time between two chromatin-binding events. Hence, 81.6% and 86.4% of CFTR-TALE and Alu-TALE molecules, respectively, are undergoing global search, whereas 18.4% of CFTR molecules are engaging in local search and 13.6% of Alu-TALE molecules are either specifically bound or are undergoing local search.

Residence time distributions of dCas9 proteins also fitted two-component exponential decay function (Fig. 2b). Our

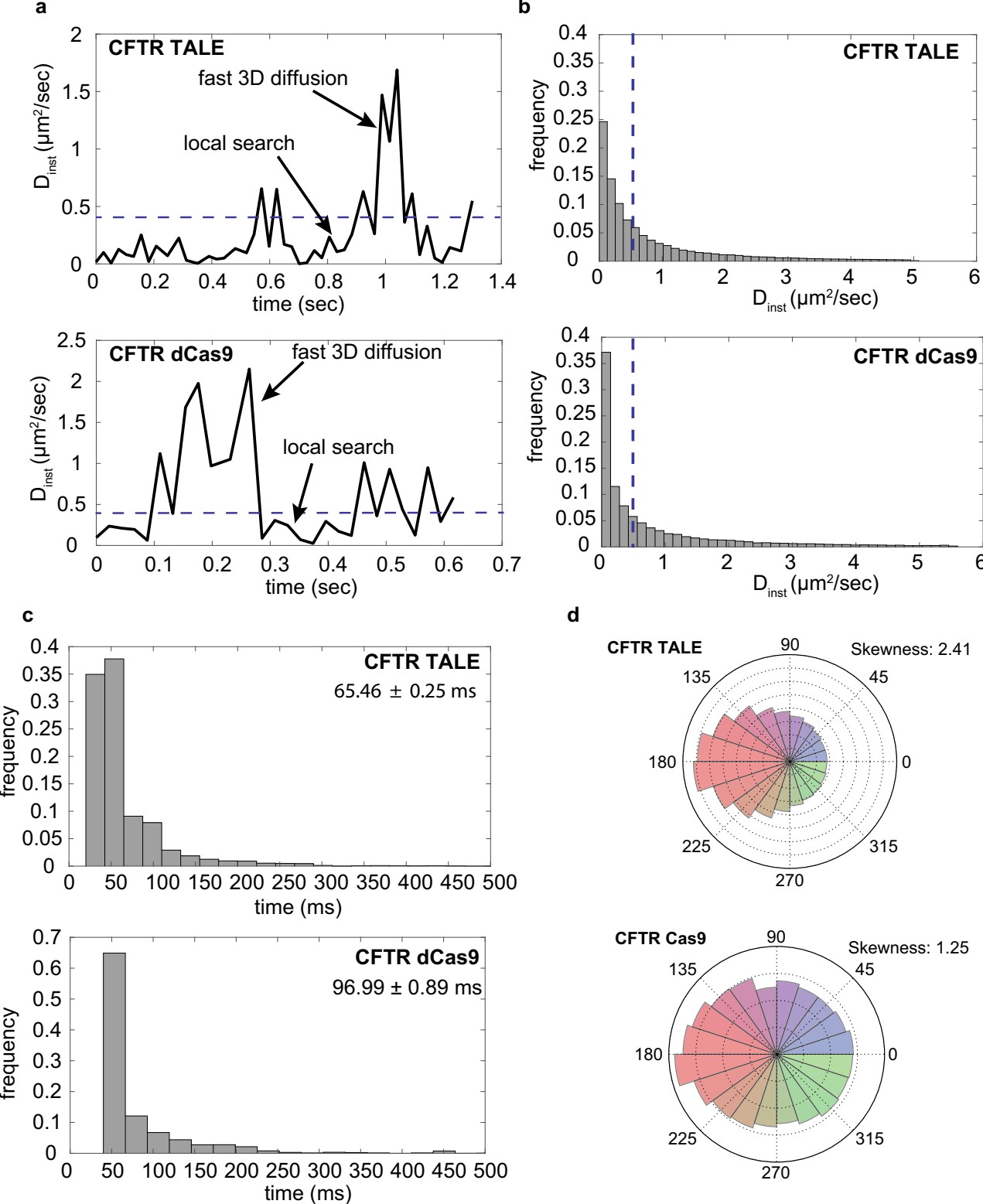

analysis shows that dCas9 spends 13.41 s on the specific target sites ($\tau_2$ of Alu-dCas9) and 5.87 s on non-specific target sites ($\tau_2$ for CFTR dCas9). dCas9 spends more time (5.87 s) on non-specific sites than TALE (1.8 sec). Since it is not possible to distinguish the target-site bound vs. non-specifically bound protein molecules with our imaging methodology, we note that the calculated residence times of specifically bound proteins could be an under-approximation as the calculation may include some of the proteins that are not specifically bound. Since the mammalian genome has more than a million Alu target sites, it is highly probable that the longer residence time corresponds to proteins that are bound to target sites, and non-specifically interacting proteins will have a minor impact on the calculated residence time. Similar analysis on H2B controls also revealed two bound populations, with long-lived populations spending 14.8 s on the genome (Supplementary Fig. 2).

**Fig. 3 Local search by TALE and dCas9. a** $D_{inst}$ of CFTR-TALE and CFTR-dCas9 is plotted with the function of time. Similar to previous in vitro studies, TALE exhibits facilitated diffusion (a combination of 3D and 1D-diffusion) in the nucleus. We observe that dCas9 is also capable of local DNA exploration. Based on the H2B control, we have defined the threshold of $D_{inst}$ (horizontal blue dashed line) that separates the locally searching and 3D diffusion proteins. **b** Plot of $D_{inst}$ of all the trajectories of CFTR-TALE and CFTR-dCas9 are shown. The dashed vertical blue line represents the threshold of $D_{inst}$. In the case of dCas9, TALE local search time is 50% of its total search time, whereas dCas9 particles indulge in local search around 56% of time approximately. $N_{CFTR\ TALE}$=114,192 and $N_{CFTR\ dCas9}$= 38,395, where N represents the number of data points. **c** dCas9 performs local search for a longer time than TALE in one local search cycle. $N_{CFTR\ TALE}$= 203,210 and $N_{CFTR\ dCas9}$= 40,170. **d** Jumping angle analysis shows that CFTR TALE is more skewed towards −180° than CFTR dCas9. The corresponding skewness factor is also shown along with the jumping angle plots. $N_{CFTR\ TALE}$= 82,867 and $N_{CFTR\ dCas9}$= 29,380. Source data are provided as a Source Data file.

**Difference between TALE and dCas9 local search behaviors**. To further understand the molecular origins of the slow-diffusing populations of TALE and dCas9, carrying out the local search, we analyzed individual trajectories of TALE and dCas9 to characterize search dynamics by calculating an instantaneous diffusion coefficient $D_{inst}$ (see "Methods"). The rapid global search was distinguished from the local search using thresholds for $D_{inst}$ (depicted by the blue dashed line in Fig. 3a)[23]. Here, $D_{inst}$ is plotted corresponding to one characteristic trajectory of a single TALE as well as a single dCas9 protein (Fig. 3a). We observed that both TALE and dCas9 proteins transition rapidly between slow and fast $D_{inst}$ ranges. TALE proteins transition between fast global and slow local search along DNA in live cells, which is consistent with prior in vitro single-molecule studies of TALE proteins[6,7]. Similarly, along with global search, dCas9 can also engage in local search of the genome in live cells.

To further analyze local search dynamics, we plotted histograms of $D_{inst}$ for TALE and dCas9 (Fig. 3b). The dashed blue vertical line demarcates the globally searching population from the locally searching population. We also determined the time spent by TALE and dCas9 in local DNA search in one cycle (Fig. 3c). Local search time per cycle is defined as the time spent by a protein molecule interacting with DNA between two consecutive cycles of global search. Our results show that the average local search fraction of dCas9 (56%) is marginally larger than that of TALE (50%). Moreover, dCas9 spends more time (96 ± 1 ms) engaging in local DNA search than TALE (65 ± 0.3 ms) per local search cycle. Overall, dCas9 spends more time undergoing local search compared to TALE.

To further probe in vivo TALE search dynamics, we determined the jumping angles of TALE proteins during the search process (Fig. 3d). Jumping angles describe the relative change in the direction of motion of a DNA-binding protein due to the local search environment encountered in the target-search process (e.g., genome compaction, other transcription factors). We also defined the skewness factor to quantify the asymmetry/non-uniformity of the jumping angle distribution (Supplementary Fig. 4). CFTR-TALE exhibits non-uniform distributions of jumping angle (skewness: 2.41), revealing that the TALE target-search process is affected by genomic occlusions (Fig. 3d). H2B proteins (Supplementary Fig. 2) demonstrate a significant bias towards 180° (skewness: 2.61), suggesting a constricted search environment. On the other hand, dCas9 shows a uniform angular distribution (skewness: 1.25) indicating that dCas9 performs efficient genome-search at the whole-nucleus level (Fig. 3d). These fundamental differences in local search efficiencies are enabling Cas9 to outperform TALENs in open chromatin. However, it is still not clear how local search will affect the performance of TALE and dCas9 in compact chromatin states of the human genome.

**TALEs navigate heterochromatin more efficiently than Cas9**. We next studied the search mechanism of TALE and dCas9 in the context of prominent genomic features in heterochromatin in live-mammalian cells. We directly imaged TALE and dCas9 search dynamics in three heterochromatin environments: Alu repetitive retrotransposons, centromeric structures, and a compact genomic locus marked by H3K9 trimethylation epigenetic modifications[24]. HeLa cells were used to image repetitive elements, and due to the availability of epigenetic data in HCT116 cells, they were used for imaging compact genomic loci. Stable (in HeLa cells) or transient (in HCT116 cells) expression of heterochromatin protein 1 alpha (HP1α) fused with the green fluorescent protein (GFP) enabled specific tracking of single protein molecules in heterochromatin[25] (Supplementary Fig. 5). Prior work has shown that TALEN and Cas9 editing activity is hindered in heterochromatin[26–28]; however, the search dynamics of these proteins in the context of heterochromatin is not understood. We observed overall slower kinetics compared to euchromatin and differential search behavior depending on the chromatin context for both dCas9 and TALE (Supplementary Fig. 6). Jumping angle analyses (Supplementary Fig. 6) demonstrate that both TALE and dCas9 encounter a considerably constricted search space as indicated by the highly skewed angular distribution in the heterochromatin region. TALE heterochromatin search is described by three distinct modes, including an additional intermediate diffusive population for repetitive elements Alu and centromere (Supplementary Fig. 6). However, when dCas9 and TALEs were designed to search for a target-site embedded in highly compacted constitutive heterochromatin located in chromosome 16, there was a significant difference between TALE (TALE 16) and dCas9 (gRNA9) search kinetics (Fig. 4a). TALE 16 ($D$ = 2.35 μm²/s) showed significantly faster overall search dynamics compared to dCas9-gRNA9 ($D$ = 1.93 μm²/s) in heterochromatin (Fig. 4a). Jumping angle distributions of TALE 16 are more uniformly distributed (skewness: 2.16) compared to that of dCas9-gRNA9 (skewness: 2.65), indicating that TALE can maneuver the tight heterochromatin environment more efficiently (Fig. 4b).

**TALEN shows higher editing efficiency than Cas9 in heterochromatin**. To assess the functional implications of differences in search behavior of TALE and dCas9 in heterochromatin, we constructed a series of TALENs and Cas9-gRNA variants capable of editing sequences present in highly repressed heterochromatin loci. Using HCT116 ENCODE H3K9me3 and H3K27me3 ChIP-seq data[29], we chose twelve chromosome loci of approximately 500 bp that differed in ChIP-seq signal fold change ranging from 2.543 to 9.547 (Supplementary Table 2). We designed four gRNAs and two TALEN pairs per loci using Benchling, CHOP-CHOP[30], and SAPTA[31] design tools (Supplementary Table 3, 4). We tested the ability of gRNA constructs to cut chromatin-less plasmid-DNA by an eGFP reporter assay where an active gRNA will cut the target-site out-of-frame with the eGFP gene resulting in loss of fluorescence upon cutting (Supplementary Fig. 7)[32]. The eGFP reporter assay enabled us to screen gRNAs that are functional and determine their editing efficiency in the context of heterochromatin (Supplementary Fig. 8). We also chose four

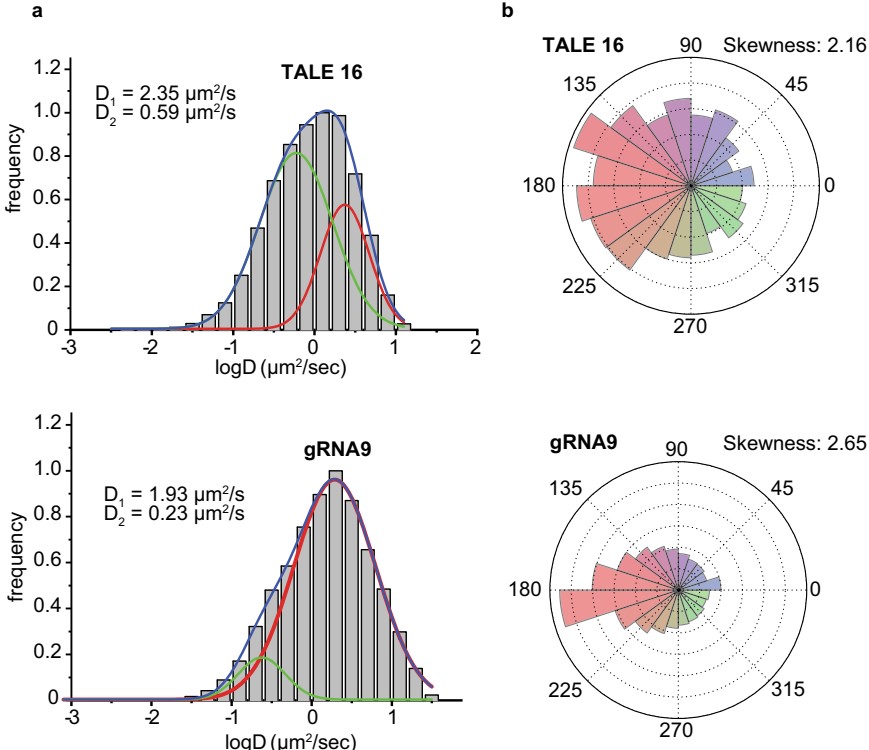

**Fig. 4 Comparison of TALE and dCas9 search dynamics in a heterochromatin locus. a** Multi-peak Gaussian fitting of normalized diffusion coefficient distribution reveals that the bimodal search behavior in heterochromatin is similar to that in euchromatin, but dCas9 kinetics is overall slower than TALE kinetics. $N_{\text{TALE 16}} = 21647$ and $N_{\text{CFTR Cas9}} = 9424$, where $N$ denotes the number of data points. **b** Pointedly skewed angular distribution curve of gRNA9 demonstrates that its motion is highly constricted in heterochromatin. $N_{\text{TALE 16}} = 1217$ and $N_{\text{CFTR Cas9}} = 22,800$. Source data are provided as a Source Data file.

published gRNAs[33] targeting euchromatin sites located in endogenous genes to compare the efficiency of genome-editing proteins in transcriptionally active open chromatin. We performed TIDE analysis[12] to calculate the target-site editing efficiency. In 11 out of 12 loci (91.66%), TALENs showed similar or higher editing activity in heterochromatin compared to Cas9 (Fig. 5a, top; Supplementary Fig. 9) whereas at 4 euchromatin sites, Cas9 demonstrated either similar or greater editing activity indicating that TALEN's enhanced editing activity in heterochromatin to be a context-dependent phenomenon (Fig. 5a, bottom). Together, the genome-editing efficiency results are consistent with in vivo search dynamics results, showing that TALE proteins are more efficient than Cas9 in navigating dense heterochromatin regions of the genome due to enhanced ability to sample heterochromatin locally. In contrast, in euchromatin, this advantage is superseded by Cas9's increased local search ability.

Based on our results, we propose a mechanistic model for the search mechanisms of TALE and dCas9 in heterochromatin that explains the difference in their relative editing performance in euchromatin and heterochromatin (Fig. 5b). Our single-molecule imaging analysis shows that not only the search mechanisms of dCas9 and TALE adapt to the chromatin environment, they also differ significantly in their extent of local search. In combination, this is correlated with their functional efficacy as Cas9 is more efficient in cutting at euchromatin sites due to its greater ability to query binding sites in a relatively unhindered environment. We hypothesize that, in heterochromatin, the enhanced local search is not a beneficial feature for Cas9 and results in reduced cutting efficiency, whereas TALEN can access heterochromatin with greater ease due to a lesser extent of local search behavior. TALE's unique rotationally decoupled DNA search mechanism[7] and short local search enable it to glide over compact heterochromatin

structures in a mammalian nucleus. On the other hand, dCas9 has to unravel the DNA double helix to interrogate for target specificity, and nucleosomes act as roadblocks for the local search, essentially trapping dCas9 molecules in the heterochromatin regions. Hence, TALE is able to find a target site embedded in mammalian heterochromatin with greater efficiency compared to dCas9.

## Discussion
In conclusion, we used a combination of single-molecule imaging and sequencing-based editing analysis to study the search dynamics of TALE and Cas9 proteins in live cells. Our results show that TALE proteins use a combination of local search and 3-D diffusion to find their target site in mammalian cells. In addition, dCas9 proteins exhibit local search behavior while sampling DNA to find the target site. We conducted a detailed single-molecule investigation of the effect of structurally distinct chromatin states on the target-search mechanism of genome-editing proteins. Alu and Centromere targeting TALEs and dCas9 variants were used to characterize the search process in prominent heterochromatin structural elements of the mammalian genome. In the case of centromeric structures, the target sites are highly repetitive and concentrated, and we observe a "hopping" like behavior of TALE and dCas9 proteins. We further show that this hopping behavior depends on the presence of similar sites in close proximity for target-searching dCas9 molecules. dCas9 targeting Alu retrotransposon elements, which are not concentrated but are interspersed throughout the genome, do not exhibit hopping behavior, which suggests that the target-search process of these proteins in heterochromatin is fundamentally different. For TALEs, the hopping behavior is seemingly

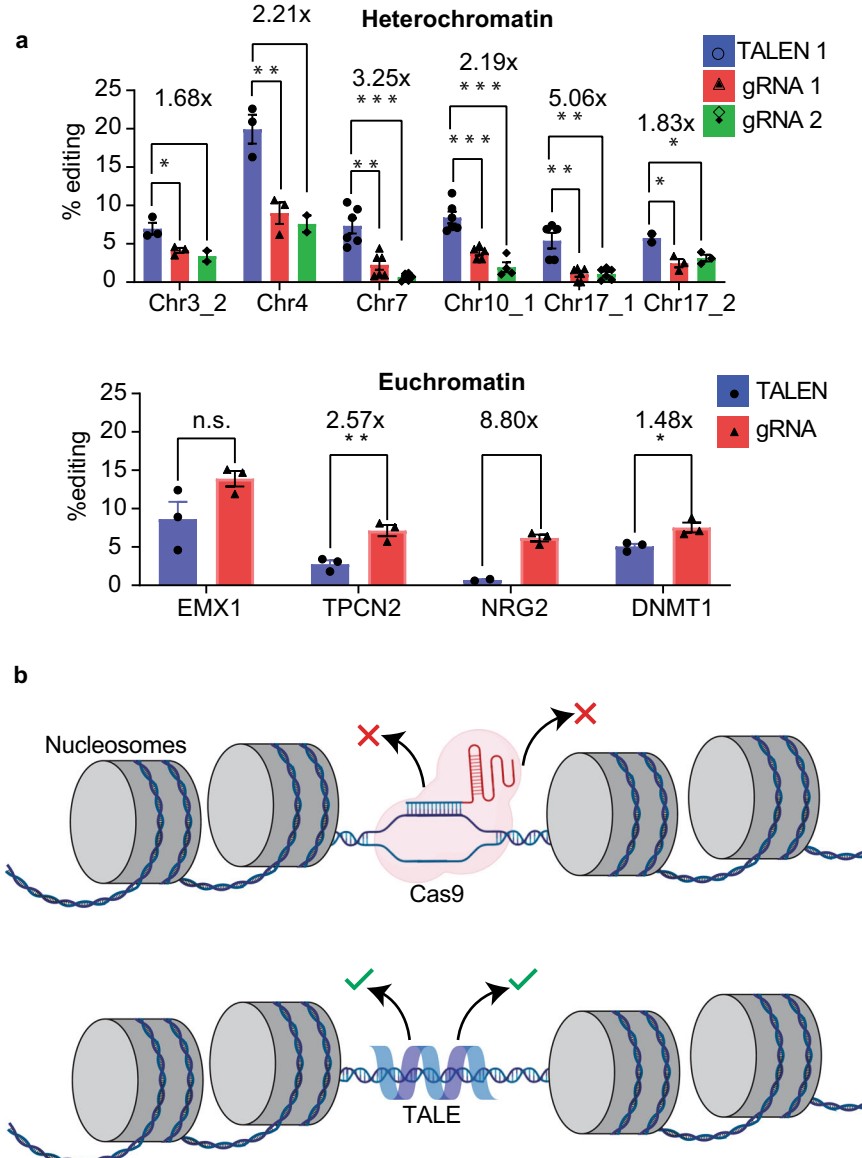

**Fig. 5 Systematic comparison of TALE and Cas9 editing efficiency in heterochromatin. a** TIDE analysis of variants of Cas9 and TALEN that target different heterochromatin and euchromatin loci. 50% of the heterochromatin loci are modified more efficiently by TALENs than Cas9 (top). Cas9 has a higher editing efficiency than TALE in the euchromatin regions (bottom). TALENs with the highest predicted cutting efficiency were designed using CHOPCHOP and SAPTA. gRNAs with the highest predicted cutting efficiency were designed using Benchling and CHOPCHOP. TALEN samples are compared to Cas9-gRNA samples by a 2-tailed $t$ test for $n > 2$. $n = \{2,3,4,5,6\}$ biological replicates. Data are presented as mean values ± SEM. $p < 0.05^*$, $p < 0.01^{**}$, $p < 0.001^{***}$. $p$-values are given in brackets: Chr3_2 (0.2755, 0.05), Chr4 (0.0096, 0.0171), Chr7 (0.0014, 6.02e-5), Chr10_1 (0.00069, 0.00033), Chr17_1 (0.00179,0.00169), Chr17_2 (0.0027,0.0326), EMX1 (0.0999), TPCN2 (0.0077), NRG2 (0.0025), DNMT1 (0.0274). Source data are provided as a Source Data file. **b** Illustration of Cas9 and TALE target-search model in heterochromatin. Cas9 becomes trapped in compact heterochromatic structures, whereas TALE can efficiently navigate the heterochromatin.

dependent on the compaction of the chromatin, but for dCas9, there is an additional requirement, perhaps the increased concentration of PAM sites or a seed-region including PAM-site.

Our results show that TALE and dCas9 search behavior is strongly dependent on the search environment and can even be of functional consequence i.e. relative genome-editing performance, as demonstrated by TIDE analysis, with TALENs emerging as the superior genome-editing tool with editing efficiencies up to 5-fold greater than that of Cas9 in heterochromatin regions. Our results show that the local search extent appears to be the most prominent distinguishing factor in determining editing outcomes at heterochromatin loci. To locate the target-site, Cas9 is dependent on local search interactions to a greater extent than TALEs, and

this becomes a disadvantage in highly compact chromatin architecture, limiting Cas9's editing efficiency in those regions. Overall, these results serve as a guide in selecting genome-editing proteins for the engineering of hard-to-edit heterochromatin regions of mammalian cells for general as well as therapeutic purposes.

## Methods
**Cell culture and transfection**. HeLa cells (ATCC® CCL-2™) were cultured in DMEM media supplemented with 10% heat-inactivated FBS (HI FBS, Life Technologies), 100 U/mL of penicillin, and 100 μg/mL streptomycin antibiotics. Cells were imaged after 24 h of transfection. HCT116 (ATCC® CCL-247™) cells were cultures in McCoy 5A medium supplemented with 10% FBS. Culture conditions were maintained at 37 °C with 5% $CO_2$. Cells were transfected using Lipofectamine

2000 (Life Technologies) or Fugene HD transfection reagent (3:1 Reagent:DNA ratio) (Promega #E2311) with plasmid DNA.

**Lentiviral transduction**. Lentiviral particles were produced in HEK293T (ATCC® CRL-3216™) cells using Fugene HD (Promega #E2311) for the transfection of plasmids. HEK293T cells were split to reach a confluency of 50–60% at the time of transfection. Lentiviral vectors were co-transfected with the lentiviral packaging plasmid psPAX2 (Addgene #12260) and the VSV-G envelope plasmid pMD2.G (Addgene #12259). Transfection reactions were assembled in reduced serum media (Opti-MEM; Gibco #31985-070). For lentiviral particle production on 6-well plates, 1 μg lentiviral vector, 0.5 μg psPAX2, and 0.25 μg pMD2.G were mixed in 0.4 mL Opti-MEM. After 15–20 min of incubation at room temperature, the transfection reactions were dispersed over HEK293T cells. The media was changed 24 h post-transfection, and the virus was harvested at 60 h post-transfection. Viral supernatants were filtered using 0.45 μm cellulose acetate or polyethersulfone (PES) membrane filters and stored at −80 °C. Polybrene (8 μg/mL; Sigma-Aldrich) was supplemented to enhance transduction efficiency.

**Plasmid construction**. TALE(N)s were assembled using an in house robotic liquid handling system Fluent following the protocol described previously[15,32]. Assembled TALEs were validated by restriction digestion (SpeI/BamHI) and Sanger sequencing with primers: N2-end-seq-F: 5′ AGCTGGATACCGGCCAACTCTT and C01-SEQ-R: 5′ ACCAGGTGGTCGTTTGTCAA. Halotag gene was further subcloned into correct TALE assemblies by Gibson assembly. Plasmid pcDNA-dCas9-Halotag expressing SpdCas9-Halotag was constructed by replacing VP64 in pcDNA-dCas9-VP64[34] purchased from Addgene (Addgene plasmid 47107) by Halotag, which was PCR amplified from custom pCMV-TALECFTR-Halotag plasmid. pcDNA-dCas9-VP64 was digested by AscI, and AflII and PCR amplified Halotag was assembled with the digested backbone into the final plasmid construct by Gibson assembly. All Gibson assemblies were carried out by using the Gibson assembly master mix (NEB #E2611L). H2B-GFP[35] was a gift from Geoff Wahl (Addgene plasmid#11680). Plasmid pcDNA-H2B-Halotag was made by assembling a PCR amplified Halotag fragment with pCMV-H2B-EGFP digested by AgeI-HF and NotI-HF. For lentiviral production, lentiv4 empty backbone harboring puromycin selection marker was digested with BamHI and AgeI, and PCR amplified GFP-HP1a[26] (Addgene #17652) was inserted into the backbone by Gibson assembly and validated by Sanger sequencing.

**gRNA design and cloning**. All gRNAs were cloned into pSPgRNA[34] featuring a U6 promoter and *Streptococcus pyogenes* gRNA scaffold was purchased from Addgene (Addgene plasmid 47108). A 20 bp guide sequence was cloned into pSPgRNA by annealing and phosphorylating two complementary oligonucleotides 5′-caccg N$_{20}$-3′ and 5′-aaacN$_{20}$c-3′, then ligating into a BbsI digested pSPgRNA backbone. N$_{20}$ represents the 20 bp guide sequence.

**Labeling and live-cell imaging**. Cells were washed with 1× phosphate buffer saline (PBS) and incubated with 2 nM of JF549 dye[18] for 15 min. Cells were washed with PBS 3x and incubated for 15 min in phenol red-free DMEM media. Finally, cells were additionally washed for 3X with PBS and plated in 35 mm glass-bottom dishes (Cellvis) in phenol red-free DMEM media. Fluorescence microscopy was performed on a Nikon Ti Eclipse microscope with ×150 magnification (CFI Apo TIRF ×100 Oil, N.A. 1.49, Nikon) using Nikon Elements software. Live cells were imaged at 30 °C in a temperature-controlled chamber (InVivo Scientific). 561 nm excitation laser (MLC400B, Agilent Technologies) was used to excite the fluorophore. A quad-band dichroic (ZT405-488-561-640RPC, Chroma) and 600/50 emission filter (Semrock) was used. An EMCCD camera (iXon DU-897E, Andor) was used to capture images at 20 ms and 500 ms exposure time. We could achieve localization accuracy up to ~5 nm for short-exposure time movies. This length scale of the localization accuracy was much shorter than the diffusion length scales of TALE and Cas9 proteins in the live-cell nucleus.

For imaging the heterochromatin region, HP1 protein was fused with GFP and was stably expressed in HCT116 cell line. GFP was imaged by 488 nm laser with the emission filter of 510/20 on the same microscope. Immediately after GFP illumination, proteins (labeled with JF646)[18] were tracked by illuminating the same area with the red laser (640 nm). Later during the analysis, using Fiji[36], we created the region of interest (ROI) based on GFP fluorescence. The same ROI was overlaid on the corresponding JF646 movie, and trajectories within the heterochromatin ROI were analyzed.

Live-cell imaging movie of CFTR fused with JF549 in HeLa cells is available in Supplementary Movie 1.

**Single-particle tracking calculations**. Movies obtained from the microscope were analyzed with Trackmate plugin[37] of Fiji[36] and trajectories were extracted. For analyzing the fast-moving trajectories, a cut-off ($D_{fast}$) of 5 μm²/s was selected. The maximum possible displacement between two frames ($R_{fast}$) was calculated from the $D_{fast}$ and was used to link particles in two consecutive frames. Trajectories were further analyzed with msdanalyzer[38] and in house written MATLAB scripts for extracting the diffusion coefficients and residence times of TALE proteins. We calculated the diffusion coefficient using an unbiased covariance-based estimator

(CVE)[39]:

$$D_{CVE,x} = \frac{(\Delta x_n)^2}{2\Delta t} + \frac{\Delta x_n \, \Delta x_{n+1}}{\Delta t}, \tag{1}$$

$$D_{CVE,y} = \frac{(\Delta y_n)^2}{2 \, \Delta t} + \frac{\Delta y_n \, \Delta y_{n+1}}{\Delta t}, \tag{2}$$

$$D_{CVE} = \frac{D_{CVE,x} + D_{CVE,y}}{2}, \tag{3}$$

where $\Delta x_n$ is $x_{n+1} − x_n$ in a trajectory time series. In this equation, $\underline{\dots}$ denotes averages over the time series $\Delta x_{1\dots}\Delta x_n$. $\Delta t$ is the exposure time.

In our 500 ms exposure time movies, fast-moving populations of proteins were blurred and we only observed the bound proteins. For these slow trajectories, diffusion coefficient cutoff ($D_{slow}$) was selected as 0.05 μm²/s. Trajectories were generated based on $D_{slow}$. The residence time of proteins is estimated as the total time a protein appeared in the movie. The single appearance of a protein was also considered. The residence time was fitted as either single or double component exponential decay model depending upon the best fit, and decay rates were calculated. The General single-exponential decay model is

$$F1(t) = f \cdot e^{\frac{t}{\tau}} \quad \text{(where } \tau \text{ is the residence time).} \tag{4}$$

The general double-exponential decay model is

$$F2(t) = f_1 \cdot e^{\frac{t}{\tau 1}} + f_2 \cdot e^{\frac{t}{\tau 2}}. \tag{5}$$

In this equation, $\tau 1$ and $\tau 2$ are the residence times of two populations and f1 and f2 are the corresponding fractions.

Photobleaching rate also affects the calculation of the residence times of the bound populations. Dissociation rates are related to the photobleaching rates in the following manner:

$$k_{output} = k_{off} + k_b, \tag{6}$$

where $k_{output}$ is the rate that is obtained from fitting the exponentials, $k_b$ is the photobleaching rate, and $k_{off}$ is the dissociation rate. Residence time can be calculated by taking the inverse of the dissociation rate:

$$\tau = \frac{1}{k_{off}}. \tag{7}$$

For calculating the jumping angles, trajectories were segregated in groups of three consecutive time points. The angle between two vectors made by three points was calculated, and polar histograms were plotted by in house written MATLAB script.

**Angular distribution analysis**. In house script was written in MATLAB to calculate the jumping angles. Three consecutive points in the trajectory were chosen and the angle was calculated between two vectors formed by points 1,2 and points 2,3. This was done for all the points in every trajectory to get the jumping angles. Jumping angles were plotted with the polar histogram function of MATLAB.

**Distance threshold for SPT**. We used Trackmate[37] plugin of ImageJ to extract the trajectories of single particles. We defined the distance threshold (r) for defining the trajectory of single protein molecules. r was defined as the maximum distance that a particle can travel in consecutive frames. The value of r is dependent on the exposure time and is calculated as follows:
For 2D diffusion coefficient,

$$r = (4Dt)1/2. \tag{8}$$

For fast diffusion,

$$D = 5 \, \mu m^2/s, \tag{9}$$

$$r_{20 \, ms} = (4 * 5 * 0.02)^{1/2} = 0.632 \, \mu m. \tag{10}$$

For 500 ms, slow diffusion,

$$D = 0.05 \, \mu m^2/s, \tag{11}$$

$$r_{500 \, ms} = (4 * 0.05 * 0.5)^{1/2} = 0.32 \, \mu m. \tag{12}$$

For 500 ms, chromatin movement,

$$D = 0.0019 \, \mu m^2/s, \tag{13}$$

$$r_{500 \, ms} = (4 * 0.0019 * 0.5)^{1/2} = 0.061 \, \mu m. \tag{14}$$

**Local search analysis**. Using mean velocity filter in TrackMate[37], a threshold to include bound molecules only was set. Using the links in tracks statistics, average displacement $d_{ave}$ overall tracks was calculated:

$D_{inst}$ was defined as

$$D_{Inst} = d_{ave}^2/4t. \tag{15}$$

For H2B bound population:

$$d_{ave} = 0.10678\,\mu m;\ t = 22\,ms, \tag{16}$$

$$D_{Inst} = 0.12961\,\mu m^2/s. \tag{17}$$

$D_{inst}$ determination for local search:

$$d_{ave} = 0.16938\,\mu m, \tag{18}$$

$$D_{Inst} = 0.326\,\mu m^2/s. \tag{19}$$

$D_{inst}$ determination for 3D search:

$$d_{ave} = 0.228\,\mu m, \tag{20}$$

$$D_{Inst} = 0.592\,\mu m^2/s. \tag{21}$$

A range of $D_{inst} \pm 1\,\mu m^2/s$ was set to define bound, local search and 3-D diffusion regimes.

The threshold of $D_{inst}$ was used to analyze the individual trajectories to characterize the cycles of local search. Each cycle of the local search was defined by the time spent by the protein locally searching the DNA between consecutive 3D search regimes. We calculated the local search cycle times and plotted their histograms in Fig. 2c.

**Reporter assay cloning and transfection**. CMV-GFP-HP1a (Addgene #17652) plasmid was modified to remove GFP-HP1a sequence using *Bam*HI and *Hind*III. This backbone was used to clone TALEN or gRNA binding sites in-frame with GFP that was amplified by PCR from the same backbone. Complementary oligos Forward Primer: ctaggccaccatggtg(N_{20}NGG)cc and Reverse Primer: gatcgg(revcomp (N_{20}NGG))caccatggtggc, containing Kozak sequence and binding site with PAM were phosphorylated and annealed and then ligated to the backbone using T7 Ligase (NEB # M0318L). 0.5×10^5 cells/well were plated in a 24-well plate, 24 h before transfection. 335 ng reporter plasmid was diluted in 26 ul pre-warmed Opti-MEM along with 195 ng of either empty plasmid (no gRNA) or gRNA plasmid. 1.65 μL Fugene HD reagent equilibriated to room temperature was added to the OptiMEM-DNA solution, mixed by pipetting 15-16 times and incubated at room temperature for 15–20 min. Resulting OptiMEM-DNA-FugeneHD mixture was added dropwise to sample wells. Cells were harvested 48 h post-transfection for flow cytometry measurements.

**Flow cytometry and analysis**. Cells were trypsinized and collected after 48 h post-transfection. Collected cells were resuspended in 500 μL PBS to prepare flow cytometry samples. Samples were analyzed on the LSR II Flow Cytometer (BD Biosciences) and data analysis was performed using FCS Express 6 (Supplementary Fig. 7b). The arithmetic mean of GFP fluorescence was used to compare Cas9-gRNA samples to a reporter only control.

**Editing comparison assay**. 4 gRNAs and 2 TALEN pairs were designed for each heterochromatin loci. One gRNA and 2 TALEN-pairs were designed for euchromatin loci. The top 2 gRNAs with the highest predicted cutting efficiency were selected using benchling CRISPR design tool (https://www.benchling.com/crispr/) and CHOPCHOP gRNA design tool (https://chopchop.cbu.uib.no/). TALEN pairs were designed with CHOPCHOP and SAPTA Scoring Algorithm (http://bao.rice.edu/Research/BioinformaticTools/TAL_targeter.html). TALEN pair constructs with the highest predicted cutting efficiency were synthesized for TIDE analysis. Cas9-gRNA and TALEN pair plasmids were transfected in HCT116 cells in equimolar amounts using Fugene HD transfection reagent following manufacturer's protocol and cell samples were collected after 48 h for TIDE analysis, which corresponds to cells undergoing at least 2 cycles. This enables averaging out of the confounding factors associated with differential HDR and NHEJ efficiency pertaining to the cell cycle stage.

**Genomic PCR and DNA sequencing**. Genomic DNA from cell pellets was extracted using QuickExtract DNA solution 1.0 (Epicenter). Genomic PCR was performed using Herculase polymerase (Agilent) or KOD polymerase with primers listed in the Supplementary Table 5. The PCR products were sequenced by Sanger DNA sequencing (Genewiz or ACGT inc.).

**TIDE analysis**. Genomic PCR products were purified using gel extraction kit (Zymo research). The indel rates were analyzed by the online software (http://tide.nki.nl) using WT sequences as reference. Default parameters were used for indel analysis of CRISPR/Cas9. For TALEN editing, 20 bp sequence between the paired binding sites was used as the "guide sequence".

**Statistical analysis**. Data are shown as mean and s.e.m. All *p*-values were generated from two-tailed *t* tests using the GraphPad Prism software package (version 6.0c, GraphPad Software) or Microsoft Excel (version 15.24).

**Reporting summary**. Further information on research design is available in the Nature Research Reporting Summary linked to this article.

## Data availability
All data are available in the main text or the supplementary information text and files. Genomic loci sequence files are provided in the Source data file. Single-molecule imaging raw datasets and any other relevant data are available from the corresponding author upon request. Source data are provided with this paper.

## Code availability
The custom codes for the data analysis used in this study are available from the corresponding author upon request. Codes can also be accessed on GitHub following this link (https://github.com/sshukla101/SelvinLab/blob/master/Diffusion_Coefficient_Calculation_CVE_Estimator.m).

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

## Acknowledgements

The authors thank L. Lavis for generously providing HaloTag ligands for imaging experiments and Tarun Chhabra and Neetesh Sharma for assistance with statistical data analysis. We also thank Kai Wen Teng and Duncan Nall from Selvin lab for their help with initial imaging experiments. We would also like to thank Guanhua Xun and Emily Gaither for help with TALEN synthesis. This work was supported by the National Institutes of Health (1U54DK107965 and 1UM1HG009402 to H.Z. and NS100019 to P.R.S.) and the National Science Foundation (PHY 1430124 to P.R.S.).

## Author contributions

Conceptualization: S.J., S.S., and H.Z.; methodology: S.J. and S.S.; Matlab scripts: S.S.; experimental investigation: S.J., S.S., C.Y., Z.F., M.Z., M.L., S.T.L., X.X., S.A.; imaging data analysis: S.S., S.J., and S.A.; ChIP-seq data analysis: Y.W., writing: S.J., S.S., H.Z., C.M.S., and P.R.S.; guidance and discussion: H.Z., P.R.S., and C.M.S., supervision: H.Z. and P.R.S.

## Competing interests

The authors declare no competing interests.
