## [Peer Review File · Nature Communications]

Reviewers' comments:

Reviewer #1 (Remarks to the Author):

The authors were able to substantially improve the clarity of the manuscript through clearer explanation and presentation. However, I believe the authors have not sufficiently differentiated between dCas9 and TALE activities in heterochromatic sites. Although centromeres and Alu sites are not functionally representative as heterochromatic sites, TALE and dCas9 showed similar behaviors. The only significant difference was demonstrated for the H3K9 methylated site.

Therefore, more convincing data needs to be presented in order to justify their implied main claim, which is that the superior search behavior of TALENs in heterochromatic regions led to higher nuclease activity. I believe this can be done by either (1) providing more evidence of consistent difference in search behavior for various methylated sites, or (2) providing more robust evidence of nuclease behavior in heterochromatic and euchromatic sites, outlined in bullet points below.

- Figure 5: Although the authors described how the guides were chosen, several more examples of Cas9 cleavage in euchromatic sites using their guide design protocol showing consistently high editing will improve the confidence that the failure of Cas9 guides in heterochromatic sites is due to the chromatin structure, not careless guide design. Another option would be to show more guide activities in those heterochromatic sites (rather than just top 2).

- Figure 5: While the authors were able to explain why Cas9 has high activity in EMX1, a euchromatic site, it is unclear why TALEN is worse at targeting a euchromatic site than a heterochromatic site. Therefore, a more representative data showing TALEN activity in other euchromatic sites will show consistency of editing by TALEN regardless of chromatin structure.

- It is unclear how the guide for EMX1 (Figure 5) was screened using a GFP reporter assay described in Extended Data Fig. 7. (Also, this figure is not mentioned in the main text). Could you elaborate on the reporter assay or provide a reference?

Reviewer #3 (Remarks to the Author):

The authors provided clear answers to the comments from the reviewers and adjusted the manuscript accordingly. The revisions have also improved the flow of the manuscript. Thus, this reviewer supports publication of the manuscript.

NCOMMS-20-25978A-Z

Title: TALEN outperforms Cas9 in editing heterochromatin target sites

Response to the editor regarding reviewer comments

We wish to express our sincere gratitude to the editor and reviewers for their time and consideration of this manuscript. We have revised the manuscript substantially and included additional heterochromatin loci for editing comparison of target sites in both heterochromatin and euchromatin to address the reviewers' concerns as well as improve the overall content of the paper. In this document, we address the reviewers' concerns, which are in black text. Our comments to the reviewers are in blue text. Sections that have been copied and pasted from the new manuscript, which have been added or edited to specifically address the reviewers' comments, are in orange text.

Overall summary of changes:

We thank the editor and the two reviewers for carefully reading our manuscript and providing valuable suggestions. We believe that these suggestions significantly improved the overall quality of our manuscript. Below, we provide our point-by-point responses and highlight additional data that has been included in the manuscript. Following Reviewer#1's recommendation to increase confidence in our data, we added 7 additional heterochromatin loci and 3 euchromatin loci and our results are consistent with our previous conclusions as delineated in the updated manuscript. We added graphic representation of our reporter assay and explained the design in the main text as well as extended data.

Response to Referee #1

Reviewer #1 (Remarks to the Author):

The authors were able to substantially improve the clarity of the manuscript through clearer explanation and presentation. However, I believe the authors have not sufficiently differentiated between dCas9 and TALE activities in heterochromatic sites. Although centromeres and Alu sites are not functionally representative as heterochromatic sites, TALE and dCas9 showed similar behaviors. The only significant difference was demonstrated for the H3K9 methylated site.

Therefore, more convincing data needs to be presented in order to justify their implied main claim, which is that the superior search behavior of TALENs in heterochromatic regions led to higher nuclease activity. I believe this can be done by either (1) providing more evidence of consistent difference in search behavior for various methylated sites, or (2) providing more robust evidence of nuclease behavior in heterochromatic and euchromatic sites, outlined in bullet points below.

- Figure 5: Although the authors described how the guides were chosen, several more examples of Cas9 cleavage in euchromatic sites using their guide design protocol showing consistently high editing will improve the confidence that the failure of Cas9 guides in heterochromatic sites is due to the chromatin structure, not careless guide design. Another option would be to show more guide activities in those heterochromatic sites (rather than just top 2).

We thank the reviewer for his/her insight and agree with him/her that our initially submitted manuscript can be improved by showing editing comparison at more euchromatin sites as well as adding more gRNA constructs to our editing comparison. To address the reviewer's concerns, we have added seven more heterochromatin sites ($n_{\text{heterochromatin_loci}}=12$) and three more euchromatin sites ($n_{\text{euchromatin_loci}}=4$). In our previous submission, we selected the top 2 gRNAs from the CHOPCHOP tool. This time, we added two more gRNAs designed using Benchling, making it a total of 4 gRNAs per heterochromatin loci. We designed 2 TALEN pairs per euchromatin site using CHOPCHOP and SAPTA and compared the editing efficiency to published euchromatin gRNAs (Slaymaker, I. M. et al. 2016)

We find that at all euchromatin sites, Cas9 performed similar (1 out of 4 sites) or better (3 out of 4 sites) than TALENs. For our comparison in heterochromatin, TALENs performed similar or better than Cas9 at 11 out of 12 sites. At 6/12 sites, TALEN performed significantly better; at 5/12 sites TALEN and Cas9 performed similarly, and at only one site, Cas9 performed better.

Lines 241-245

Using HCT116 ENCODE H3K9me3 and H3K27me3 ChIP-seq data²⁹, we chose twelve chromosome loci of approximately 500 bp that differed in ChIP-seq signal fold change ranging from 2.543 to 9.547 (Extended Data Table 2). We designed 4 gRNAs and two TALEN pairs per loci using Benchling, CHOPCHOP30, and SAPTA31 design tools.

Lines 251-256

We performed TIDE analysis¹³ to calculate the target-site editing efficiency. In 11 out of 12 loci (91.66%), TALENs showed similar or higher editing activity in heterochromatin compared to Cas9 (Fig. 5a, top; Extended Data Fig. 8) whereas at 4 euchromatin sites, Cas9 demonstrated either similar or greater editing activity indicating that TALEN's enhanced editing activity in heterochromatin to be a context-dependent phenomenon (Fig. 5a, bottom).

- Figure 5: While the authors were able to explain why Cas9 has high activity in EMX1, a euchromatic site, it is unclear why TALEN is worse at targeting a euchromatic site than a heterochromatic site. Therefore, a more representative data showing TALEN activity in other euchromatic sites will show consistency of editing by TALEN regardless of chromatin structure.

We thank the reviewer for his/her thoughtful suggestion and for giving us a chance to further test our hypothesis, "Cas9's greater local search extent in euchromatin makes it a better genome-editing protein in euchromatin" by performing TIDE analysis at additional euchromatin sites. We designed 2 TALENs (predicted top candidate from CHOPCHOP and SAPTA tools) to compare with published gRNAs that target endogenous genes in euchromatin. We observed that Cas9 performs better than TALEN in euchromatin at three out of four euchromatin sites (NRG2, TPCN2, DNMT1) and similarly at one site (EMX1), pointing to Cas9 being a more efficient 'searcher' in euchromatin.

In principle, this observation is in line with the native function of these genome-editing proteins. TALENs are transcription activator-like effector proteins that have to be functional in higher eukaryotes like plants that possess a complex genome with heterochromatin. In contrast,

CRISPR/Cas9 is found in bacteria, which of course, lack specialized chromatin domains such as heterochromatin. It's possible that Cas9 did not evolve to cut in compact heterochromatin sites, while TALENs perform consistently in all chromatin domains.

However, based on our single-molecule analysis, the extent of local search determines the cutting efficiency of genome editing proteins, and search behavior is dependent on the chromatin environment as evident by the emergence of an intermediate search population for centromere dCas9 and TALE variants, possibly pertaining to proteins 'jumping' between multiple available target sites. Hence, we would like to make the case that local search is dependent on chromatin state and is directly correlated with cutting efficiency in the respective chromatin state.

Lines 254-256

...whereas at 4 euchromatin sites, Cas9 demonstrated either similar or greater editing activity indicating that TALEN's enhanced editing activity in heterochromatin to be a context-dependent phenomenon (Fig. 5a, bottom).

Lines 301-305

Our results show that the local search extent appears to be the most prominent distinguishing factor in determining editing outcomes at heterochromatin loci. To locate the target-site, Cas9 is dependent on local search interactions to a greater extent than TALEs, and this becomes a disadvantage in highly compact chromatin architecture, limiting Cas9's editing efficiency.

- It is unclear how the guide for EMX1 (Figure 5) was screened using a GFP reporter assay described in Extended Data Fig. 7. (Also, this figure is not mentioned in the main text). Could you elaborate on the reporter assay or provide a reference?

We thank the reviewer for noticing our error and allowing us to clarify our experimental details. Before conducting the TIDE analysis, we employed the eGFP reporter assay to determine if the gRNA is functional in a chromatin-free context. We designed cloning sites upstream of the eGFP start codon to insert the gRNA binding site (target site+PAM) (Extended Data Fig. 7a). We have added a description of the reporter assay in the main text and Methods section, and we have referenced Extended Data Fig 7 in the main text as well.

We have added three more euchromatin gRNAs- NRG2, TCPN2, DNMT1 along with EMX1 and compared the editing efficiency with two different TALEN pairs designed using CHOPCHOP and SAPTA tools. In Fig. 5b, we plot the best performing TALEN pair's editing efficiency and compare it with the gRNA for the corresponding gene (these gRNAs were selected from Slaymaker, I. M. et al. (2016) and are shown to be functional in their manuscript).

Lines 245-249

We tested the ability of gRNA constructs to cut chromatin-less plasmid-DNA by an eGFP reporter assay where an active gRNA will cut the target site in-frame with the eGFP gene resulting in loss of fluorescence upon cutting by an active gRNA (Extended Data Fig. 7A). The eGFP reporter assay enabled us to screen gRNAs that are functional and determine their editing efficiency in the context of heterochromatin.

Reviewer #3 (Remarks to the Author):

The authors provided clear answers to the comments from the reviewers and adjusted the manuscript accordingly. The revisions have also improved the flow of the manuscript. Thus, this reviewer supports publication of the manuscript.

We would like to take this opportunity to thank the reviewer for his/her consideration of our manuscript and would like to extend our sincere gratitude for his/her positive recommendation for publication.

REVIEWERS' COMMENTS

Reviewer #1 (Remarks to the Author):

This reviewer appreciates the steps taken to improve the overall quality of the manuscript as well as providing convincing data to support their conclusions. I support the publication of this manuscript.

(Minor typo: In line 251, it should read "out-of-frame" instead of "in-frame" in order to result in fluorescence loss.)

Response to reviewers' comments:

Reviewer #1 (Remarks to the Author):

This reviewer appreciates the steps taken to improve the overall quality of the manuscript as well as providing convincing data to support their conclusions. I support the publication of this manuscript.

(Minor typo: In line 251, it should read “out-of-frame” instead of “in-frame” in order to result in fluorescence loss.)

We appreciate this reviewer' positive comments. The typo has been fixed.